



# Reflected ray retrieval from radio occultation data using radio holographic filtering of wave fields in ray space

Michael E. Gorbunov[1], Estel Cardellach[2], and Kent B. Lauritsen[3]

[1]A. M. Obukhov Institute of Atmospheric Physics RAS, Moscow
[2]Institut d'Estudis Espacials de Catalunya, Barcelona, Spain
[3]Danish Meteorological Institute, Copenhagen, Denmark

*Correspondence to:* Michael Gorbunov (gorbunov@ifaran.ru)

**Abstract.** Linear and non-linear representations of wave fields constitute the basis of modern algorithms for analysis of radio occultation (RO) data. Linear representations are implemented by Fourier Integral Operators, which allow for high-resolution retrieval of bending angles. Non-linear representations include Wigner Distribution Function (WDF), which equal the pseudo-density of energy in the ray space. Representations allow for filtering wave fields by suppressing some areas of the ray space and mapping the field back from the transformed space to the initial one. We apply this technique to the retrieval of reflected rays from RO observations. The use of reflected rays may increase the accuracy of the retrieval of the atmospheric refractivity. Reflected rays can be identified by the visual inspection of WDF or spectrogram plots. Numerous examples from COSMIC data indicate that reflections are mostly observed over oceans or snow, in particular, over Antarctica. We introduce the reflection index that characterizes the relative intensity of the reflected ray with respect to the direct ray. The index allows for the automatic identification of events with reflections. We use the radio holographic estimate of the errors of the retrieved bending angle profiles of reflected rays. A comparison of indices evaluated for a large base of events including the visual identification of reflections indicated a good agreement.

## 1 Introduction

A clear signature of signals reflected by the Earth's surface was revealed as early as the beginning of 21st century, by means of the radio holographic analysis of CHAMP radio occultation (RO) data (Beyerle and Hocke, 2001; Beyerle et al., 2002). Similar patterns were also found in Microlab-1 GPS/MET data (Gorbunov, 2002b, c). It was pointed out that the utilization of reflected signals can be useful for the enhancement of the retrievals. Reflections are mostly observed above water (ocean) or snow (Antarctica). Another application of reflected signals is linked to the altimetry (Cardellach et al., 2004).

Currently, the main means of identification of reflections remains the radio holographic analysis in sliding apertures (Gorbunov, 2002c; Cardellach et al., 2009, 2010), which has also been used to extract the reflected fields (Cardellach et al., 2015; Aparicio et al., 2017). Alternatively, the reflected signals can also be identified with techniques based on the Wigner Distribution Function (WDF) (Gorbunov et al., 2010, 2012). However, it is known that the techniques based on different approximations for the Fourier Integral Operator, developed for retrieval of RO in multipath areas, are also capable of retrieving the reflected part of the bending angle (BA) profile. These techniques include: Canonical Transform (CT) methods based on



the concept of Fourier Integral Operatos (FIO) (Gorbunov, 2002a; Gorbunov and Lauritsen, 2004), Full Spectrum Inversion (FSI) (Jensen et al., 2003), and Phase Matching (PM) (Jensen et al., 2004).

In this paper, we discuss the algorithm of reflected ray retrieval based on the modification of the CT method. The algorithm is based on the filtering in ray space. In the CT method, the original wave field observed as a function of observation time $t$ is projected into the impact parameter domain. The field in the transformed space is a function of impact parameter $p$, and the direct and reflected rays can be classified according to their value of $p$. The reflected rays have the impact parameter value below that corresponding to the geometric optical shadow. Therefore, the reflected field component can be separated in the impact parameter domain. For the further processing, it is more convenient to project the reflected field component back into the time domain (Cardellach et al., 2009, 2010). For this projection we use the inverse FIO. Generally, our approach is similar to that developed in (Cardellach et al., 2009, 2010), where the spectrograms were filtered in order to remove direct rays and inverted. However, the CT-based approach allows for constructing efficient numerical algorithms. In the problem of the reflected ray retrieval, this approach has the same advantages of the radio holographic (sliding spectral) method (Beyerle and Hocke, 2001; Beyerle et al., 2002) as in the problem of the direct ray retrieval: it is more accurate and compuatationally more efficient (Gorbunov, 2002c). This work was published as a technical report (Gorbunov, 2016).

The paper is organized as follows. In Section 2, we discuss possible approaches to the reflection retrieval. We present typical examples of reflections observed in COSMIC data, which allow us to choose the most adequate method of reflection retrieval. In Section 3 we discuss the phase model for reflected rays, and its use for the definition of the reflection index that is a measure of the reflection strenght. Next we discuss the filtering algorithm based on FIO that allow separating reflection from RO observations. Section 4 shows a few examples of COSMIC events with different reflection indices. In Section 5, we make our conclusions.

## 2 Possible Approaches to Reflection Retrieval

In this Section, we will discuss possible retrieval algorithms for the retrieval of reflected rays. As compared to direct rays, reflected rays are characterized by a lower amplitude and by a bending angle profile rapidly increasing with impact height. This makes it difficult to apply the CT algorithm directly. Below we discuss three algorithms: 1) direct application of CT technique, 2) the modified CT technique where the impact parameter is replaced with a linear combination of impact parameter and bending angle, and 3) composition of a filter in the impact parameter space that suppresses the direct rays with subsequent geometric optical inversion in the time domain.

### 2.1 Reflected Rays Retrieval in Impact Parameter Domain

The measured wave field has the following form:

$$u(t) = A(t) \exp\left(ik\left(S_0(t) + \Delta S(t)\right)\right), \tag{1}$$





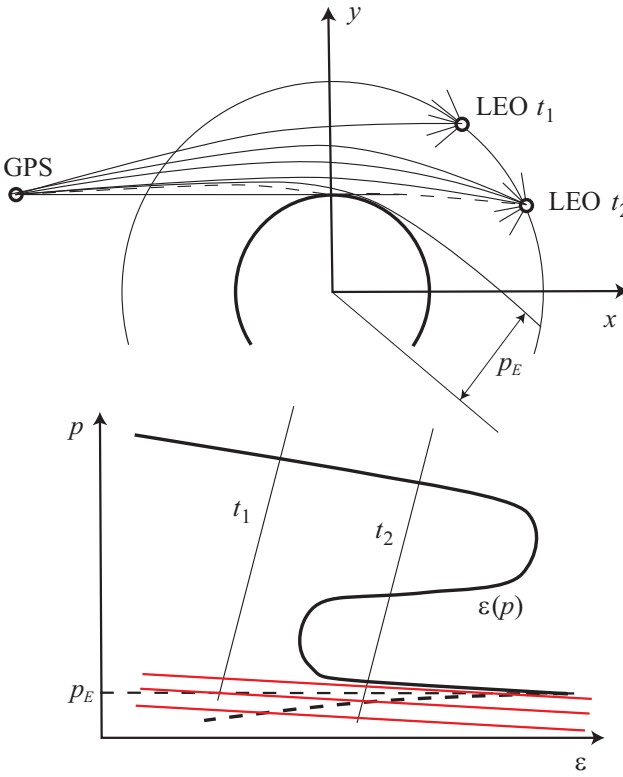

**Figure 1.** Reflection and multipath propagation geometry. Upper panel shows the radio occultation geometry. At moment of time $t_1$, one ray is observed, at moment of time $t_2$ the receiver is in a multipath area and observes 3 direct rays and one reflected ray. (lower panel) Each moment of time corresponds to a line in the impact parameter – bending angle coordinate plane; the line indicating possible values of impact parameters and bending angles that can be observed by the receiver at this moment of time. $p_E$ is the impact parameter of the ray that touches the Earth's surface. Reflected rays have impact parameters $p < p_E$. The bending angle profile of reflected rays is shown as a dashed line. In red are shown the lines of constant modified impact parameter.

where $A(t)$ is the observed amplitude, $S_0(t)$ is the satellite-to-satellite distance evaluated from the orbit data, $\Delta S(t)$ is the observed atmospheric excess phase. The smoothed phase excess $\overline{\Delta S}(t)$ and its derivative $\overline{\Delta S'}(t)$ are obtained by the filtering of the measured phase excess $\Delta S(t)$. The smoothed relative Doppler shift $\bar{d}(t)$ is obtained as follows:

$$\bar{d}(t) = d_0(t) - \frac{\overline{\Delta S'}(t)}{c}, \tag{2}$$

5   where $d_0(t) = S'_0(t)/c$ is the vacuum relative Doppler frequency shift. From $\bar{d}(t)$ and satellite orbit data we evaluate the smooth impact parameter model $\bar{p}(t)$, bending angle model $\bar{\varepsilon}(t)$, and the derivative of impact parameter over Doppler shift $d\bar{p}(t)/d\bar{d}$ (Vorob'ev and Krasil'nikova, 1994). The ancillary function $f(t)$ is evaluated as follows (Gorbunov and Lauritsen, 2004):

$$f(t) = \bar{p}(t) - \bar{d}(t)\frac{d\bar{p}(t)}{d\bar{d}}. \tag{3}$$



The new coordinate is determined as follows (Gorbunov and Lauritsen, 2004):

$$\Upsilon(t) = \Upsilon_0 - c \int_{t_0}^{t} \left( \frac{d\bar{p}(t)}{d\bar{d}} \right)^{-1} dt', \tag{4}$$

where $\Upsilon_0$ is a constant determined in such a way that $\Upsilon(t) \geq 0$ for the observation time interval. We evaluate the integral of $f(t)$:

$$f_I(t) = \int_{\Upsilon(t_0)}^{\Upsilon(t)} f(t') \, d\Upsilon(t'). \tag{5}$$

Using this function, we evaluate the vacuum and observed phase path with subtracted model as follows:

$$S_0^{(M)}(t) = S_0(t) - R_E \Upsilon(t) + f_I(t), \tag{6}$$
$$S^{(M)}(t) = S_0^{(M)}(t) + \Delta S(t), \tag{7}$$

where $R_E$ is the Earth's local curvature radius. Subtraction of $R_E \Upsilon(t)$, a linear function of $\Upsilon$, from the phase corresponds to

the reduction of the frequency, which equals the impact parameter, by a constant of $R_E$. All the functions of time $t$ can also be looked at as functions of the new coordinate $\Upsilon$.

The Fourier Integral Operator is defined as follows (Gorbunov and Lauritsen, 2004):

$$\hat{\Phi}_2 u(\tilde{p}) = \sqrt{\frac{k}{2\pi}} a(\tilde{p}) \int A(\Upsilon) \exp\left( ikS^{(M)}(\Upsilon) - ik\tilde{p}\Upsilon \right) d\Upsilon, \tag{8}$$

where $a(p)$ is the amplitude function, whose definition can be found in (Gorbunov and Lauritsen, 2004). The variable $\tilde{p}$ is the

approximate impact height (impact parameter with subtracted $R_E$ due to the definition of $S_0^{(M)}(t)$). The transformed field $\hat{\Phi}_2 u(\tilde{p})$ is represented as follows:

$$\hat{\Phi}_2 u(\tilde{p}) = A'(\tilde{p}) \exp\left( i\varphi'(\tilde{p}) \right), \tag{9}$$

where $A'(\tilde{p})$ is the amplitude of the transformed field and $\varphi'(\tilde{p})$ is its accumulated phase. The frequency variable $\Upsilon$ is defined in such a way that it is always positive in the area, where any rays may be expected. This simplifies the evaluation of the

accumulated phase $\varphi'(\tilde{p})$. The amplitude function is evaluated using $\varphi'(\tilde{p})$, which is the reason why field $u(\tilde{p})$ is first evaluated up to this factor.

The analysis of the amplitude of the field in the transformed space allows for the determination of the shadow border impact height $\tilde{p}_E$ (Gorbunov, 2002a; Gorbunov and Lauritsen, 2004; Jensen et al., 2004), as shown in Figure 1. Practically, because the energy of reflected rays is much smaller than that of direct rays, this will also be the border between direct and reflected rays.

The CT2 algorithm evaluates the filtered phase derivative, separately for $\tilde{p} < \tilde{p}_E$ and for $\tilde{p} > \tilde{p}_E$, which we denote as follows:

$$\frac{d\bar{\varphi}'_D(\tilde{p})}{d\tilde{p}}, \quad \frac{d\bar{\varphi}'_R(\tilde{p})}{d\tilde{p}}, \tag{10}$$





where subscript D stays for direct rays, and subscript R stays for reflected rays. However, it will be necessary to implement an additional option that specifies the filter width for reflected rays. This is explained by the fact that the impact parameter interval for reflected rays is usually as narrow as 100–200 m. This requires a narrow filter window of about 20 m, while the typical setting for processing direct rays in the lowest troposphere is 250 m.

5 ## 2.2 Reflected Rays Retrieval in Modified Impact Parameter Domain

In order to circumvent the problem of the reflected ray retrieval in the impact parameter space, caused by steep increase of bending angle profile for reflected rays, we consider the following modification of the CT method.

The FIO (8) corresponds to the following linear canonical transform (Gorbunov and Lauritsen, 2004):

$$\tilde{p} = f(\Upsilon) + \eta, \tag{11}$$

$$\xi = -\Upsilon, \tag{12}$$

where $\eta$ is the eikonal derivative (momentum) of the original observed field $u(t)$, and $\xi$ is the momentum of the transformed field. This transform can be modified in order to use another coordinate:

$$\tilde{p}' = \tilde{p} + \beta\Upsilon, \tag{13}$$

where $\beta$ is a tunable parameter. Lines of constant coordinate $\tilde{p}'$ are shown in Figure 1 in red. This indicates that the bending 15 angle profile of reflected ray is less steep in this coordinate space for $\beta > 0$, because:

$$\frac{d\varepsilon}{d\tilde{p}'} = \frac{\frac{d\varepsilon}{d\tilde{p}}}{1 + \beta\frac{d\varepsilon}{d\tilde{p}'}}. \tag{14}$$

On the other hand, $\beta$ cannot be made too large, because in this case, the direct rays may overlap with the reflected rays in the modified space.

The modification of the integral transform is staightforward. The modified canonical transform is written as follows:

20 
$$\tilde{p}' = f(\Upsilon) + \beta\Upsilon + \eta \equiv f'(\Upsilon) + \eta, \tag{15}$$

$$\xi = -\Upsilon. \tag{16}$$

Using the modified function $f'(\Upsilon)$ instead of the original one defined in (3), we obtain the expression for the modified FIO $\hat{\Phi}'_2$. The advantage of this approach is that it can be implemented by a relatively small modification of the existing CT2 algorithm. Its disadvantage is the presence of a tunable parameter $\beta$, whose optimal value is unkonwn in advance and may vary from event 25 to event.

### 2.3 Reflected Rays Retrieval in Time Domain

The CT2 algorithm is designed for the retrieval of bending angle profiles in multipath areas, where the profiles are non-monotonic. This is not the case for bending angle profiles of reflected rays, which always monotonically increase. This makes




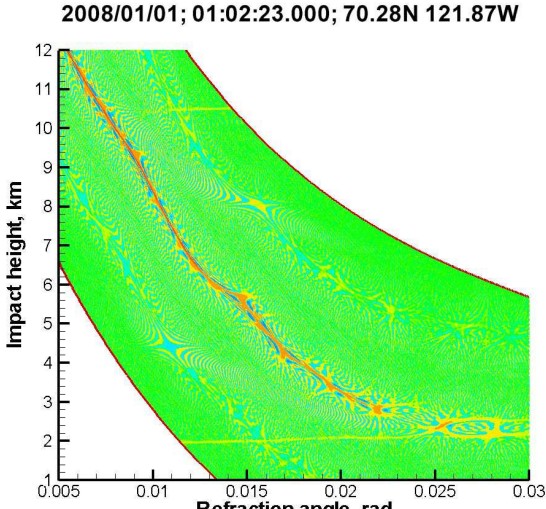
**2008/01/01; 01:02:23.000; 70.28N 121.87W**

**Figure 2.** Reflection over ice/snow.

it convenient to retrieve the dependence $p(\varepsilon)$ rather than $\varepsilon(p)$. On the other hand, it is more straightforward to formulate the retrieval algorithm in the time domain, as illustrated by Figure 1. In the time domain, in presence of reflection, there is always multipath propagation due to the interference of direct and reflected rays. But the field component related to the direct rays can be effectively removed. To this end, we can use the impact parameter domain, where the direct and reflected rays are clearly

separated by the border impact height of $\tilde{p}_\mathrm{E}$. Therefore, we can form the following field $u_R(t)$ in the time domain that only contains the reflections:

$$u_R(t) = \hat{\Phi}_2^{-1}\left[\hat{\Phi}_2\left[u(t)\right]\theta\left(\tilde{p}_E - \tilde{p}\right)\right], \tag{17}$$

where $\hat{\Phi}_2$ is the FIO, $\hat{\Phi}_2^{-1}$ is its inverse, and $\theta$ is the theta-function, which takes the value of 1 for positive arguments and 0 for negative arguments. This function can then be processed using the standard geometric optical (GO) technique. The advantage

of this approach is that it is free of tunable parameters, and the final retrieval of reflected rays is performed in the time domain, which is the optimal coordinate for the manifold of reflected rays. This is illustrated by Figure 1, which shows a very typical bending angle profile of reflected rays, where at each moment of time only one reflected ray is observed. This will also be illustrated by examples of experimental data.

## 2.4   Examples of Reflections in COSMIC Observations and Discussion

Figure 2 through Figure 5 show examples of reflections detected in COSMIC observations. Each Figure includes the event time, location, and the 2-D plot of the WDF for the observed wave field (Gorbunov et al., 2010) (cf. Figure 1). Reflections are observed over ocean or snow/ice. Many interesting examples are observed over Antarctica. See (Aparicio et al., 2017) for a statistical analysis of the reflection events.


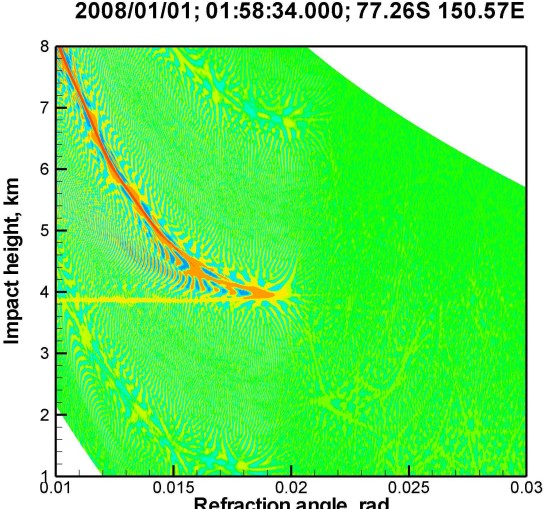

**Figure 3.** Reflection over ice/snow in Antarctica.

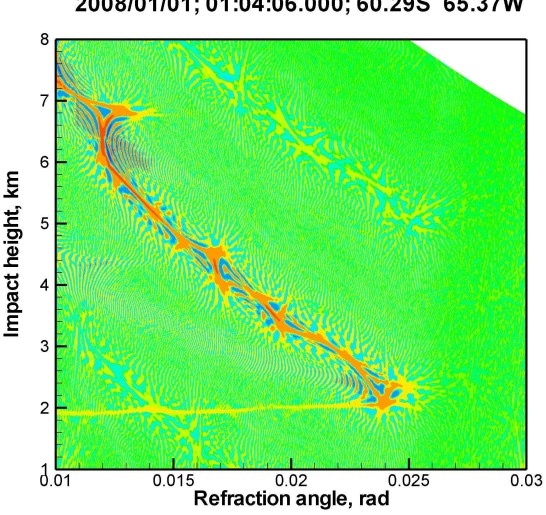

**Figure 4.** Reflection over ocean.

The examples of reflections allow for the following conclusion. Observed reflections indicate a very rapid increase of the bending angle of reflected rays, $\varepsilon_R$ as a function of impact parameter $p$. Dependence $\varepsilon_R(p)$ is mostly confined in a narrow impact parameter interval of about 100 m. Often $\varepsilon_R(p)$ is a multi-valued function. This indicates the method of choice should



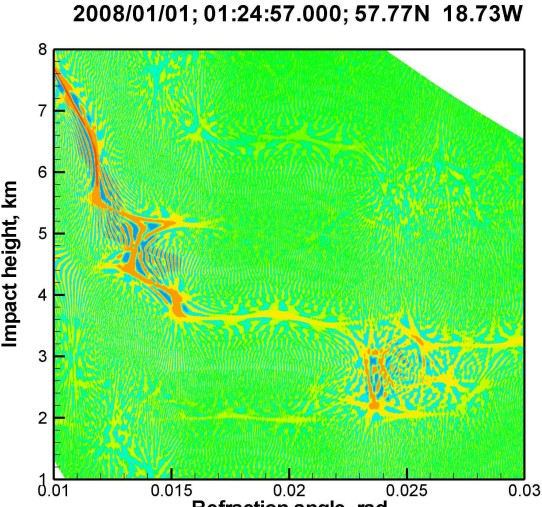

**Figure 5.** Reflection over ocean.

be the time domain retrieval preceded by the extraction of the reflected signal by using the filtering in the impact parameter space.

## 3 Reflection Retrieval Implementation

### 3.1 Phase Model for Reflected RO Signals

The phase model will play an important role in our further discussion. Here we describe the algorithm for the evaluation of the phase model for reflected RO signals. Given a spherically symmetric model of the neutral atmosphere $n_M(r)$, where $r$ is the distance from the Earth's curvature center, the corresponding bending angle profile for reflected rays is expressed as follows (Cardellach et al., 2015; Aparicio et al., 2017):

$$\varepsilon_{MR}(p) = -2p \int\limits_{p_E}^{\infty} \frac{d \ln n_M}{dx} \frac{dx}{\sqrt{x^2 - p^2}} - 2 \arccos\left(\frac{p}{p_E}\right), \qquad (18)$$

where $x(r) = r n_M(r)$ is the refractive radius, the first term describing the refraction due to the refractivity gradient, and the second term describing the ray bending due to the reflection at the surface, $p_E = r_E n_M(r_E)$, and $r_E$ is the Earth's curvature radius with the account of the surface height above the reference ellipsoid. Our neutral atmospheric model $n_M(r)$ is based on the MSIS-90 model complemented with 80% relative humidity below 15 km as described in (Gorbunov et al., 2011). An example of reflected bending angle model $\varepsilon_{MR}(p)$ is shown in Figure 6. Together with the satellite orbit data, the model

bending angle profile $\varepsilon_{MR}(p)$ allows for the determination of the phase excess for the reflected rays. To this end, we have first



to numerically solve the following equation:

$$\theta(t) = \varepsilon_{MR}(p) + \arccos \frac{p}{r_{\text{Tx}}(t)} + \arccos \frac{p}{r_{\text{Rx}}(t)}, \tag{19}$$

where $\theta(t)$ is the satellite-to-satellite angle with respect to the local curvature center, $r_{\text{Tx,Rx}}(t)$ are the radial coordinates of the satellites, hereinafter index Tx staying for the transmitter and index Rx staying for the receiver. The equation is solved for time

$t$ for each prescribed impact parameter. This allows for the determination of impact parameters as function of time, $p_{MR}(t)$. Dependence $p_{MR}(t)$ is always single-valued for reflected rays, because reflected bending angle profiles are monotonic and do not result in multipath propagation. This is illustrated by Figure 1 and explained by eq. (18), where the derivative of the second, reflective term proves to be much stronger than that of the first, refractive term, for any possible atmospheric conditions. Given satellite coordinates $\mathbf{x}_{\text{Tx,Rx}}(t)$, the ray directions at the satellites, unit vectors $\mathbf{u}_{\text{Tx,Rx}}(t)$ are inferred from $p(t)$ using the

geometrical relationships:

$$\mathbf{x}_{\text{Rx}}(t) \times \mathbf{u}_{\text{Rx}}(t) = \mathbf{x}_{\text{Tx}}(t) \times \mathbf{u}_{\text{Tx}}(t), \tag{20}$$

$$|\mathbf{x}_{\text{Rx}}(t) \times \mathbf{u}_{\text{Rx}}(t)| = |\mathbf{x}_{\text{Tx}}(t) \times \mathbf{u}_{\text{Tx}}(t)| = p, \tag{21}$$

which express the fact that rays lie in the vertical occultation plane, and the impact parameter has the same value at the transmitter and at the receiver. Using the satellite velocities $\mathbf{V}_{\text{Rx,Tx}}(t)$, we find the relative Doppler frequency shift $d_{MR}(t)$

$$\mathbf{V}_{\text{Tx}}(t) \cdot \mathbf{u}_{\text{Tx}}(t) - \mathbf{V}_{\text{Rx}}(t) \cdot \mathbf{u}_{\text{Rx}}(t) = c d_{MR}(t). \tag{22}$$

The phase excess is obtained by integrating the Doppler shift:

$$S_{MR}(t) = c \int \left( d^{(0)}(t) - d_{MR}(t) \right) dt, \tag{23}$$

where $d^{(0)}(t)$ is the vacuum Doppler shift for the direct rays, evaluated from (40), by inserting unit vector $\mathbf{u}_{\text{Tx,Rx}}^{(0)}(t)$ corresponding to satellite-to-satellite straight-line direction. An example of reflected phase excess model is shown in Figure 7.

## 3.2   Radio Holographic Index of Reflections

The idea of flagging radio occultation with an index of the strength of the reflected ray consists in the following. Although the amplitude of the reflected signal is weak as compared to the direct signal, the instant frequencies of the reflected signal concentrate around the instant frequencies of the model reflected signal. Therefore, we can use the model reflected signal $\exp(ik S_{MR}(r))$ as the reference signal and evaluate the radio holographic spectrum as follows:

$$\tilde{u}_R(\omega) = \int A(t) \exp(ik [S(t) - S_{MR}(t)] - i\omega t) dt. \tag{24}$$

As the reference signal, we use the smoothed reflected signal phase excess $\overline{S_R}(t)$ rather than the model $S_{MR}(t)$. We apply the sliding polynomial smoothing with a window of about 1 s. This modification makes the radio holographic spectrum sharper, while its maximum for reflection is located closer to the zero frequency. The integration here covers the time interval, for which





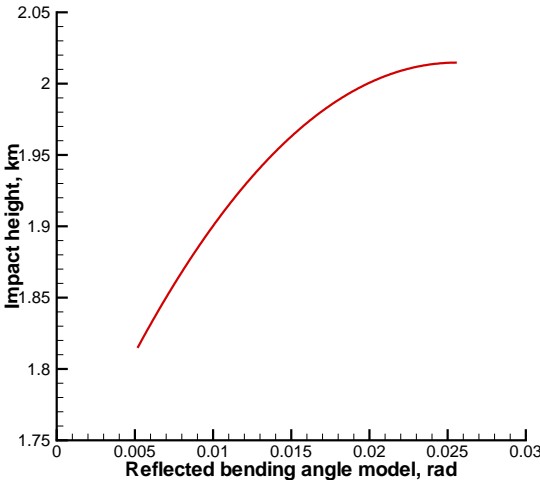

**Figure 6.** Reflected bending angle model for occultation event 2008/01/01, UTC 01:02:23, 70.28°N 121.87°W (the same event as in Figure 2).

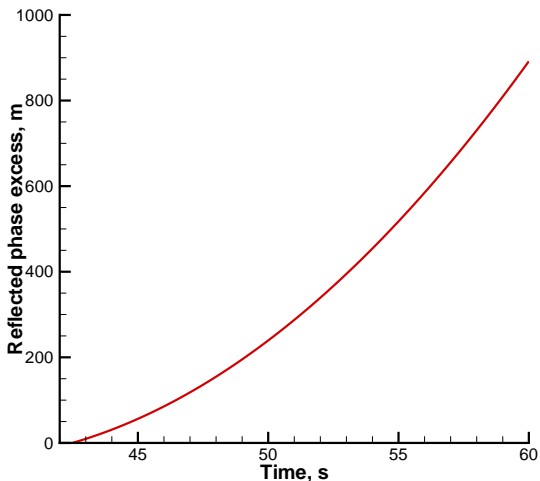

**Figure 7.** Reflected phase excess model for occultation event 2008/01/01, UTC 01:02:23, 70.28°N 121.87°W (the same event as in Figure 2).

we can evaluate the reflected phase excess model. Each frequency $\omega$ can be transformed to the equivalent impact parameter. This allows for considering the spectrum as a function of impact parameter related to middle point $t_0$ of the time interval. Moreover,





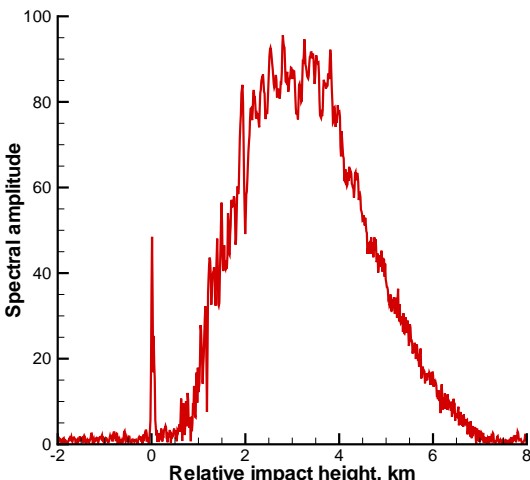

**Figure 8.** Radio holographic spectrum amplitude $|\tilde{u}_R(\omega)|$ for occultation event 2008/01/01, UTC 01:02:23, 70.28°N 121.87°W.

it is convenient to introduce the reference value $p_0$ of the impact parameter, corresponding to frequency $\omega_0 = \dot{S}_{MR}(t_0)$. The spectrum can be considered as function $\tilde{u}_R(\Delta p)$ of relative impact height $\Delta p = p - p_0$.

An example of radio holographic spectrum is shown in Figure 8. The spectrum indicates a distinct spike near $\Delta p = 0$, corresponding to reflection. The presence of reflection is also confirmed by the Wigner function plot in Figure 2.

Using the radio holographic spectrum, we define the reflection index $I_R$ as follows:

$$\tilde{u}_{\max} = \max_{[-0.1,\,0.1]} |\tilde{u}(\Delta p)|^2, \tag{25}$$

$$\tilde{u}_{\text{ave}} = \left\langle |\tilde{u}(\Delta p)|^2 \right\rangle_{[p_{\max}-0.3,\,p_{\max}+0.3]}, \tag{26}$$

$$\tilde{u}_{\text{bkg}} = \left\langle |\tilde{u}(\Delta p)|^2 \right\rangle_{[1.0, 2.0]}, \tag{27}$$

$$I_R = \frac{\tilde{u}_{\max}^2}{\tilde{u}_{\text{ave}}(\tilde{u}_{\max} + \alpha\,\tilde{u}_{\text{bkg}})} \left\langle \exp\left(-\left[\frac{p(t) - p_M(t)}{2\delta p(t)}\right]^2\right) \right\rangle, \tag{28}$$

where $\tilde{u}_{\max}$ is the maximum of the spectral density taken within the interval of $\Delta p \in [-0.1\text{ km},\ 0.1\text{ km}]$, $p_{\max}$ is the location of the spectral maximum of the reflection, $\tilde{u}_{\text{ave}}$ is the spectral density averaged over the interval of $[p_{\max} - 0.3, p_{\max} + 0.3]$, $\tilde{u}_{\text{bkg}}$ is the background (noise level) spectral density estimated by averaging over the interval of $\Delta p \in [1.0\text{ km}, 2.0\text{ km}]$, and $\alpha$ is the regularization parameter, $p_M(t)$ is the dependence of the impact parameter on the model reflected signal versus time, and $\delta p(t)$ is the radio holographic error estimate of the impact parameter. The regularization allows for suppressing random
maxima at the noise level, if the reflection is weak or absent. We estimate the background spectrum density $\tilde{u}_{\text{bkg}}$ from the impact parameter interval of $[1.0, 2.0]$ km, where a signal from direct ray is present. The regularization strength is controlled by the parameter $\alpha$: reflection is only identified if $\tilde{u}_{\max}$ significantly exceeds both $\tilde{u}_{\text{ave}}$ and $\alpha\,\tilde{u}_{\text{bkg}}$. The optimal value of $\alpha$ was




empirically estimated to be about 0.2. The additional exponential factor in the definition of $I_R$ penalizes profiles deviating too much from the model. The averaging in this factor is spread over the whole domain, where the reflected bending angle profile is evaluated. This reflection index definition is easy to implement and computationally inexpensive.

The index characterizes the strength of the spectral spike and suppresses random spikes at noise level. The value of $I_R = 0.25$

corresponds to a flat radio holographic spectrum, i.e. a definite absence of reflection. For the illustrative event considered above, the index has a value of 17.917. This index can therefore be used to identify presence of reflected signals in RO data.

### 3.3 Filtering in Impact Parameter Space

In order to extract the reflected field component $u_R(t)$, we implemented the filtering in the impact parameter space. In Figure 2, we see that the reflected ray is observed both around impact height of 2 km and 10.5 km. The latter originates from aliasing,

where the Doppler frequency shift of the reflected ray deviates from the direct ray phase excess model by more than a half of the receiver band width, which equals 50 Hz. The impact parameters difference $\Delta p_{\text{alias}}$ between non-aliased and aliased components for typical observation geometry is about 8–10 km. The exact value $\Delta p_{\text{alias}}$ for a specific event is evaluated by using GO relationship (21) and (22). To this end, we evaluate impact height from the original relative Doppler shift, and from the relative Doppler shift corresponding to the aliased frequency shifted by the sampling rate. In order to retain the aliased

component, we modify filter (17) as follows:

$$u_R(t) = \hat{\Phi}_2^{-1}\left[\hat{\Phi}_2\left[u(t)\right]\chi(\tilde{p})\right], \tag{29}$$

where $\chi(\tilde{p})$ is equal to unity inside the impact height interval of $[\tilde{p}_E - \Delta p_R, \tilde{p}_E]$ and the corresponding aliased interval of $[\tilde{p}_E + \Delta p_{\text{alias}} - \Delta p_R, \tilde{p}_E + \Delta p_{\text{alias}}]$. The width $\Delta p_R$ of these intervals is set to 1 km. Outside these intervals, we employ the Gaussian apodization with a characteristic width of $\delta p_R = 0.2$ km. Apodization allows for avoiding sharp boundaries of the

filtering function, improving the filter quality. For the estimate of the impact parameter uncertainty, we employ the radio holographic analysis. (Gorbunov et al., 2006).

### 4 Examples of Processing COSMIC Data

Figures from 9 to 12 show a few examples of processing COSMIC data. The examples start with a strong reflections, demonstrate further event with decreasing reflection index, and last showing an event with no reflection. These examples demonstrate

that the reflection index can serve as a measure of reflected ray strength.

In order to validate our retrieval algorithm and reflection index definition statistically, we performed a comparison of our retrievals with the ROM SAF reflection flag database (ROM SAF, 2016). The database contains occultation events classified into three categories: 1) no reflection, 2) reflection, and 3) unclear. The events are accompanied by the SVM (Supporting Vector Machine) index (Cardellach et al., 2009, 2010) based on the radio holographic analysis and supervised learning method.

The reflection index is a functional of a process containing a random component (noise, turbulence effects etc.) and deterministic regular structures (direct and reflected ray). The index characterizes the intensity and the sharpness of the reflected



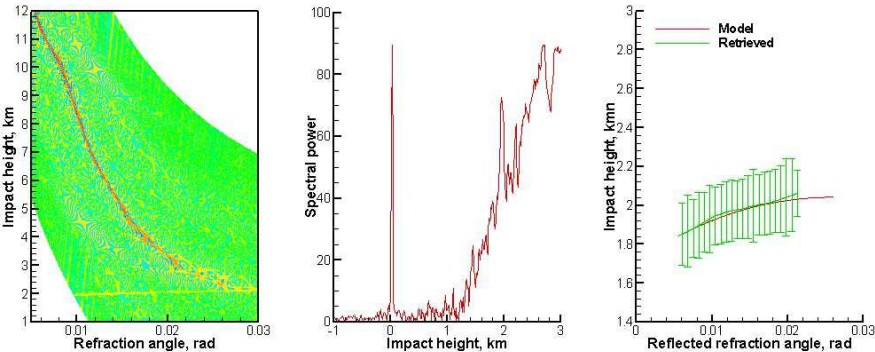

**Figure 9.** Occultation event 2008/01/01, UTC 14:59:50, 72.34°N 164.30°W. Left: Wigner distribution function; middle: radio holographic spectrum; right: model and retrieved reflection bending angles, the latter with error bars indicating the uncertainty estimate. Reflection index 27.240.

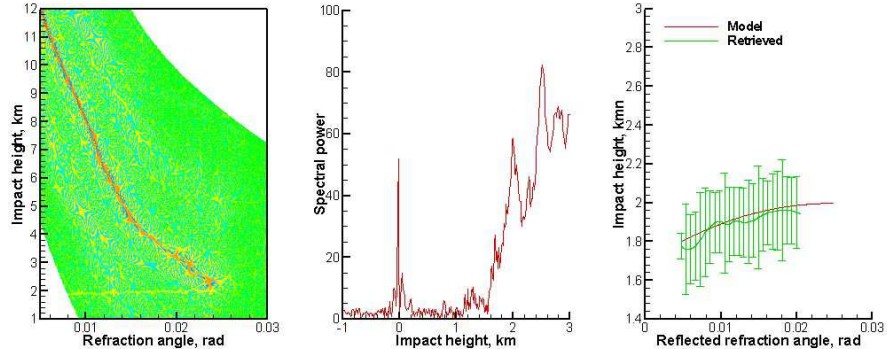

**Figure 10.** Occultation event 2008/01/01, UTC 08:46:37, 61.38°S 69.50°E. Left: Wigner distribution function; middle: radio holographic spectrum; right: model and retrieved reflection bending angles, the latter with error bars indicating the uncertainty estimate. Reflection index 20.755.

ray. Being a functional of a random process, the index is itself a random quantity with its own distribution. None index, under these conditions, can exactly characterize the regular structure in 100% cases. Instead, it characterize the probability of reflection occurrence. Practically, the use of the index is accompanied by setting a threshold. The events with the index below the threshold are rejected, the remaining events are treated as those containing reflection. The higher the threshold is chosen, the

5 higher is the probability, and the less events will pass the threshold. Practically the threshold is chosen from the comparison of the index with the visual investigation of an ensemble of events that is large enough for providing statistically significant results.





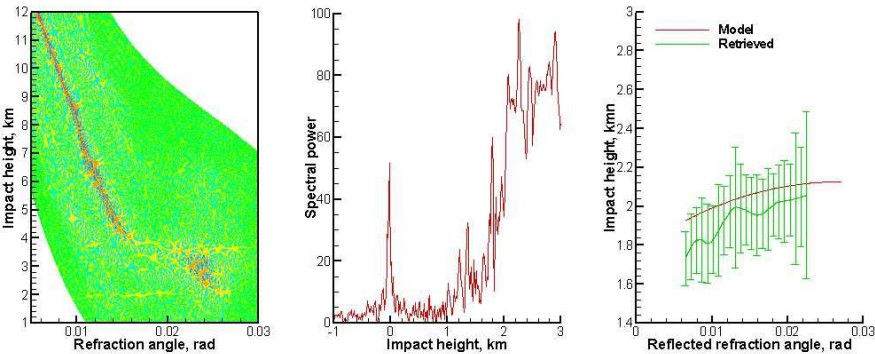

**Figure 11.** Occultation event 2008/01/01, UTC 10:07:11, 48.68°S 134.46°W. Left: Wigner distribution function; middle: radio holographic spectrum; right: model and retrieved reflection bending angles, the latter with error bars indicating the uncertainty estimate. Reflection index 13.617.

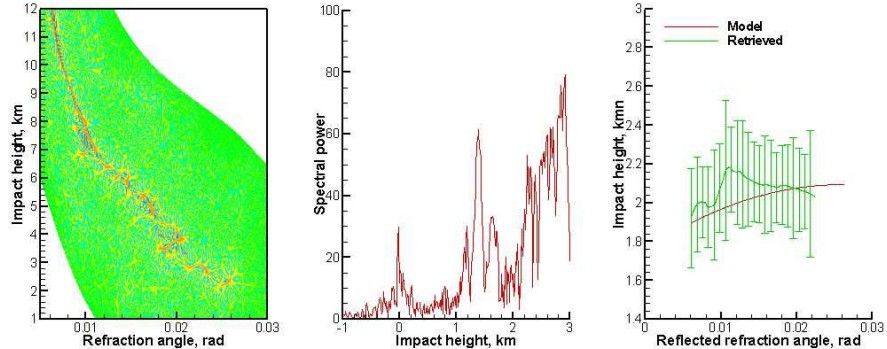

**Figure 12.** Occultation event 2008/01/01, UTC 04:39:47, 44.78°N 44.75°W. Left: Wigner distribution function; middle: radio holographic spectrum; right: model and retrieved reflection bending angles, the latter with error bars indicating the uncertainty estimate. Reflection index 8.104.

Figure 14 shows the probability distribution function (PDF) of the reflection index $I_R$ for the three categories of events. For the no-reflection category, the PDF has a strong maximum for events with the index below 1. For the reflection cases, the PDF has a tail for indexes below 3. At $I_R = 3$, the PDFs for no-reflection and reflection cases have an equal magnitude. This allows for taking the value of 3 as a lowest threshold. About 5% of events classified as clear reflection will be rejected by 5 this threshold. At $I_R = 5$, the PDF of no-reflection cases reaches 0. This allows for taking the value of 5 as the highest (safe) threshold. About 10% of events classified as clear reflection will be rejected by this threshold.





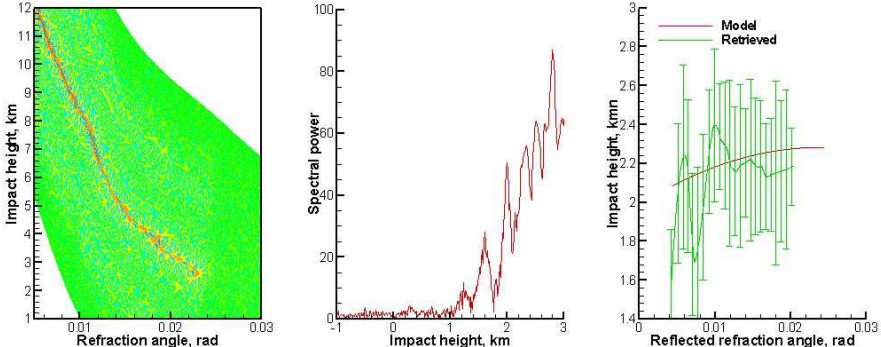

**Figure 13.** Occultation event 2008/01/01, UTC 02:51:36, 49.16°N 76.55°W. Left: Wigner distribution function; middle: radio holographic spectrum; right: model and retrieved reflection bending angles, the latter with error bars indicating the uncertainty estimate. Reflection index 0.072.

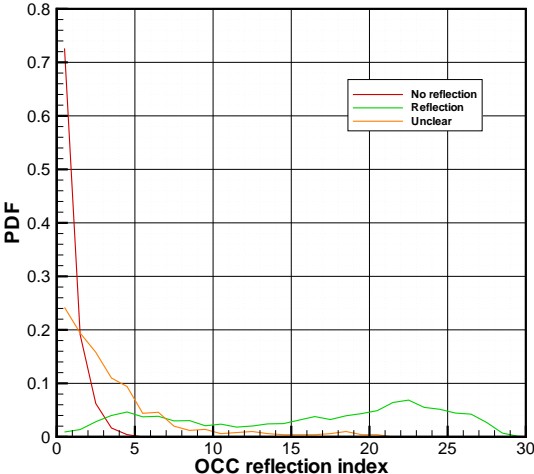

**Figure 14.** Probability distribution function of $I_R$ for the three categories of events.

## 5   Conclusions

In this paper, we described our modification of the CT technique for the retrieval of bending angle profiles of reflected rays. Our approach uses the combination of the filtering in the impact parameters space with the standard GO retrieval. The filtering uses the FIO in order to map the observed wave field to the impact parameter space. The field in the transformed space is multiplied

5   with the filter function which suppresses the direct ray and let only pass both the not aliased and aliased components of the reflected ray. The filtered field is mapped back to the time domain. The phase of the resulting field is re-accumulated in the





vicinity of the phase model of the reflected ray. We use the radio holographic spectra in order to estimate the reflection index and the expected error of the impact parameter. The reflection index characterizes the strength of the reflection. We validated our reflection index definition by a comparison with the ROM SAF reflection flag database. In general, our reflection index indicates a good agreement with the database. Some discrepancies are partly explained by the misclassification of some events

in the database, and partly by the random nature of RO signals resulting in an overrated reflection index for some tropical events. These events are located on the distribution tails. Based on this comparison, it is possible to estimate the threshold values of the reflection index. Its values exceeding 5 allow for speaking about a definite presence of reflection. Its values below 3 are typical for the absence of reflection. Values between 3 and 5 may correspond to different cases with likely or unlikely reflection. The extracted profiles of reflected bending angle and impact parameter have potential to be assimilated into NWP

models through the forward operator in Equation 17 (Aparicio et al., 2017). They might contribute anchoring the atmospheric conditions at the surface level. Studies are being conducted within the EUMETSAT ROM SAF to assess their added value and impact in NWP assimilation scenarios.

## 6   Code availability

A public version of the code used for this study is being prepared for the ROM SAF Radio Occultation Processing Package

(ROPP); http://www.romsaf.org/ropp/

## 7   Data availability

The data used can be sent by request.

*Acknowledgements.* Part of this work (Section 2 and part of Section 3) was conducted as part of the Visiting Scientist program of the Radio Occultation Meteorology Satellite Applications Facility (ROM SAF), which is a decentralized operational radio occultation processing center

under EUMETSAT. M.E.G. was a ROM SAF Visiting Scientist for the CDOP-2 VS27 project and E.C. and K.B.L. are members of the ROM SAF. The work on the rest of Section 3 and Section 4 was co-supported by Russian Foundation for Basic Research (grant RFBR No. 16-05-00358 A).





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
