# Peer review of "Reflected ray retrieval from radio occultation data using radio holographic filtering of wave fields in ray space"

_Atmospheric Measurement Techniques, 2017_

## Referee Comment (RC1) · Anonymous Referee #1 · 6 Sep 2017

The authors present an advanced method to detect reflection signals in GPS occultation data. The method is clearly described with nice viewgraphs, detailed formulas and examples. I do not notice any error. Generally, the article is well suited for a publication in the AMT journal.
* * *

---

## Referee Comment (RC2) · Anonymous Referee #2 · 23 Nov 2017

This paper presents a new approach in the retrieval and identification of reflected signals in RO data using radioholographic (RH) techniques. It is this reviewer's belief that the use of reflected signal is currently being underutilized, and this paper represents a step forward in the right direction. Overall, I find the paper technically interesting, although at times unfocused, with some concepts/motivations not well explained.

(1) The WDF method was used throughout the paper to illustrate the presence/absence of reflection in the bending angle/impact parameter space. However, the paper uses a CT-based technique to retrieve the reflected signal. It is not clear why the authors could not simply use WDF. Please explain.

[Figure]

(2) Section 2.1, p.5: The authors wrote that the "impact parameter interval for reflected rays is usually as narrow as 100-200 m. This requires a narrow filter window of about 20 m, while the typical setting for processing direct rays in the lowest troposphere is 250 m." This served as the motivation for a modified impact parameter described in Section 2.2. However, I do not understand why this is a problem in practice. What exactly is the problem of using a narrow filter window?

(3) Section 2.2, p.5: The modified impact parameter approach was introduced here and almost immediately discarded because "the presence of the tunable parameter beta." If this method is not useful, why did the authors bother to introduce it here at all?

(4) Section 2.3, p.5: "The CT2 algorithm is designed for the retrieval of the bending angle profiles in multipath areas, where the profiles are non-monotonic. This is not the case for bending angle profiles of reflected rays, which always monotonically increase." The explanation for the second statement came much later, in p. 9. I suggest either alerting the readers that the explanation will come later or moving the explanation up. Now about that explanation. . ..

(5) p. 9: ". . . explained by eq. (18), where the derivative of the second, reflective term proves to be much stronger than that of the first, refractive term, for any possible conditions." I'd like to see a proof of that, using more realistic atmospheric conditions than modified MSIS. Close to the surface, the direct and reflected rays are almost merging. If multipath can affect direct rays, couldn't it also affect reflected rays?

(6) Section 2.3: My understanding is that the RH approach was ultimately only used to filter out the direct signal (eq. 17) and then transform back to the time domain. This begs the question of whether RH is really necessary. Can you show an example where the identification using sliding spectrum in the time domain is problematic but solved using the RH filtering approach?

(7) Section 2.4: "Figure 2 through Figure 5 show examples of reflections detected in COSMIC observations." There were very few discussions of the individual figures.

First, the reflection features in these figures should be clearly marked. Second, provide more details on what differentiates these figures. If they are not sufficiently different, please consider eliminating some of them.

(8) p. 7: "Often epsilon_R(p) is a multi-valued function." Doesn't that contradict the earlier statement that it "always monotonically increase"?

(9) p. 8: "r_E is the Earth's curvature radius with the account of the surface height above the reference ellipsoid." Do you mean the height over the geoid or terrain height?

(10) Eq. (24), p. 9: Please define A(t) and S(t). Are these the amplitude and phase of the complex signal that has already gone through the RH filtering of Eq. (17). If so, why is the spectral amplitude so large at positive impact parameters?

(11) The reflection index depends on a number of subjective parameters (Eqs 25-28). Please quantify sensitivity of the reflection index on these parameters. Giving the value of reflection index in such high precision (e.g., 20.755) seems misleading.

(12) Figs. 9-13: Please describe how the uncertainties are derived and what they mean. They are so large that they seem consistent with no reflections?

(13) The RH formulation presented assumes spherical symmetry. What if the atmosphere is spherically symmetric but the surface is not? In Beyerle et al., there was discussion of how surface tilts will affect the reflected Doppler shift. Could you comment on how a surface tilt will affect your analysis?

Minor corrections:

(14) Throughout: "phase excess" should be replaced by "excess phase"

(15) Above Eq. (24), p. 9: "S_{MR}(r)" -> "S_{MR}(t)"

(16) p. 13: "None index" -> "No index"

---

## Author Response (AR1)

*The authors present an advanced method to detect reflection signals in GPS occultation data. The method is clearly described with nice viewgraphs, detailed formulas and examples. I do not notice any error. Generally, the article is well suited for a publication in the AMT journal.*

We are grateful to the reviewer for the positive estimate of our paper.

**Anonymous Referee #2**

*This paper presents a new approach in the retrieval and identification of reflected signals in RO data using radioholographic (RH) techniques. It is this reviewer's belief that the use of reflected signal is currently being underutilized, and this paper represents a step forward in the right direction. Overall, I find the paper technically interesting, although at times unfocused, with some concepts/motivations not well explained.*

*(1) The WDF method was used throughout the paper to illustrate the presence/absence of reflection in the bending angle/impact parameter space. However, the paper uses a CT-based technique to retrieve the reflected signal. It is not clear why the authors could not simply use WDF. Please explain.*

The reason is that WDF is a good method for the visualization of RO data, but it is not that accurate in the bending angle retrieval, as discussed by Gorbunov et al., 2012. A corresponding remark was added to the text.

*(2) Section 2.1, p.5: The authors wrote that the "impact parameter interval for reflected rays is usually as narrow as 100-200 m. This requires a narrow filter window of about 20 m, while the typical setting for processing direct rays in the lowest troposphere is 250 m." This served as the motivation for a modified impact parameter described in Section 2.2. However, I do not understand why this is a problem in practice. What exactly is the problem of using a narrow filter window?*

Such a filter will not be able to effectively suppress random noise. This explanation has been added in the manuscript, top of page 5.

*(3) Section 2.2, p.5: The modified impact parameter approach was introduced here and almost immediately discarded because "the presence of the tunable parameter $\beta$." If this method is not useful, why did the authors bother to introduce it here at all?*

During this study, this method, as one of possible solutions, was implemented and tested. The method was found to work. The further analysis showed that the method based on the RH filter has strong advantages. Still, we anticipate that the modified canonical transform with the tunable parameter $\beta$ may be useful in the development of advanced retrieval algorithms, so we decided to describe it.

*(4) Section 2.3, p.5: "The CT2 algorithm is designed for the retrieval of the bending angle profiles in multipath areas, where the profiles are non-monotonic. This is not the case for bending angle profiles of reflected rays, which always monotonically increase." The explanation for the second statement came much later, in p. 9. I suggest either alerting the readers that the explanation will come later or moving the explanation up. Now about that explanation....*

We added a reference to the later discussion of the monotonicity of bending angle profiles of reflected rays.

*(5) p. 9: "... explained by eq. (18), where the derivative of the second, reflective term proves to be much stronger than that of the first, refractive term, for any possible conditions." I'd like to see a proof of that, using more realistic atmospheric conditions than modified MSIS. Close to the surface, the direct and reflected rays are almost merging. If multipath can affect direct rays, couldn't it also affect reflected rays?*

It is hardly possible to give a general mathematical proof of that for an arbitrary medium, but it is straightforward to give an estimate for realistic conditions. Because strong multipath effects are caused by superrefraction layers, and they are the strongest for spherically layered medium, we can write:

$$\varepsilon_R\left(p\right) = -2p\int_{p_E}^{\infty}\frac{d\ln n}{dx}\frac{dx}{\sqrt{x^2 - p^2}} - 2\arccos\left(\frac{p}{p_E}\right)$$

$$\approx -\sqrt{2r_E}\int_{r_E}^{\infty}\frac{d\ln n\left(r\right)}{dr}\frac{dr}{\sqrt{rn\left(r\right) - p}} - 2\arccos\left(\frac{p}{p_E}\right),$$

Then we can write the expression for the derivative of the bending angle:

$$\frac{d\varepsilon_R\left(p\right)}{dp} \approx \sqrt{\frac{r_E}{2}}\int_{r_E}^{\infty}\frac{d\ln n\left(r\right)}{dr}\frac{dr}{\left(rn\left(r\right) - p\right)^{3/2}} + \sqrt{\frac{2}{r_E}}\frac{1}{\sqrt{p_E - p}},$$

Now, assuming that the strongest perturbation comes from a superrefraction layer with a thickness of $\Delta r$, critical refractivity gradient of $-r_E^{-1}$ and located at an altitude of $h_{SR}$, we can write:

$$\frac{d\varepsilon_R\left(p\right)}{dp} \approx -\sqrt{\frac{1}{2r_E}}\frac{\Delta r}{h_{SR}^{3/2}} + \sqrt{\frac{2}{r_E}}\frac{1}{\Delta p^{1/2}} = \sqrt{\frac{1}{2r_E h_{SR}}}\left(-\frac{\Delta r}{h_{SR}} + 2\sqrt{\frac{h_{SR}}{\Delta p}}\right),$$

where $\Delta p = p_E - p$. Assuming that $h_{SR}$ is about the PBL height, i.e. 1.5 km, and the superrefraction layer thickness is about 0.2 km, and $\Delta p < 0.2$ km, we see that $\Delta r / h_{SR} \ll 2\sqrt{h_{SR}/\Delta p}$ and, therefore, $d\varepsilon_R\left(p\right)/dp > 0$. Because the bending angle profile of reflected rays is monotonic, there is only one ray at each moment of time, as illustrated by Figure 1. This estimate agrees with our experience of data analysis. We look at hundreds of plots and never saw multipath propagation effects for reflected rays, which indicates that conditions that break the above assumptions occur rarely, if at all. The above estimate was added to the text.

*(6) Section 2.3: My understanding is that the RH approach was ultimately only used to filter out the direct signal (eq. 17) and then transform back to the time domain. This begs the question of whether RH is really necessary. Can you show an example where the identification using sliding spectrum in the time domain is problematic but solved using the RH filtering approach?*
RH is really necessary, because it removes the direct ray overlapping with the reflected ray in the time domain. After the removal of the direct ray, it becomes possible to apply the geometric optical retrieval technique for the reflected ray retrieval.

*(7) Section 2.4: "Figure 2 through Figure 5 show examples of reflections detected in COSMIC observations." There were very few discussions of the individual figures.*
*First, the reflection features in these figures should be clearly marked. Second, provide more details on what differentiates these figures. If they are not sufficiently different, please consider eliminating some of them.*
Figure 1 provides an explanation how the reflection feature should look like. In the caption of Figure 2 we added the following explanation: "The branch of bending angle profile corresponding to the reflection looks like a nearly horizontal line at the impact height of about 2 km. Cf. Figure 1." We prefer keeping all these Figures because 1) they serve as good illustration of different conditions resulting in reflection and 2) the paper has a reasonable volume.

*(8) p. 7: "Often $\epsilon_R\left(p\right)$ is a multi-valued function." Doesn't that contradict the earlier statement that it "always monotonically increase"?*
We agree that some additional explanations are necessary. The statement that $\epsilon_R\left(p\right)$ relates to a hypothetical case of spherically symmetric medium. Horizontal gradients may result in multi-valued $\epsilon_R\left(p\right)$.

1. Gorbunov, M. E. & Kornblueh, L. (2001), 'Analysis and validation of GPS/MET radio occultation data', *Journal of Geophysical Research* **106**(D15), 17,161-17,169.

2.  Gorbunov, M. E. & Lauritsen, K. B. (2009), Error Estimate of Bending Angles in the Presence of Strong Horizontal Gradients, *in* A. Steiner; B. Pirscher; U. Foelsche & G. Kirchengast, ed., 'New Horizons in Occultation Research', Springer, Berlin, Heidelberg, pp. 17--26.

*(9) p. 8: "$r_E$ is the Earth's curvature radius with the account of the surface height above the reference ellipsoid." Do you mean the height over the geoid or terrain height?*
We mean the geoid. The text is corrected.

*(10) Eq. (24), p. 9: Please define $A(t)$ and $S(t)$. Are these the amplitude and phase of the complex signal that has already gone through the RH filtering of Eq. (17). If so, why is the spectral amplitude so large at positive impact parameters?*
$A(t)$ and $S(t)$ are the amplitude and excess phase of the observed signal before the RH filter. This information has now been added in the manuscript.

*(11) The reflection index depends on a number of subjective parameters (Eqs 25-28). Please quantify sensitivity of the reflection index on these parameters. Giving the value of reflection index in such high precision (e.g., 20.755) seems misleading.*
These parameters are not "subjective", because they reflect objective physical properties of the signal, noise and observation geometry. However, these parameters are not strictly derived, and, therefore, they are "empirical", as we stated in the text.
The values of these parameters were optimized during the study. Currently, the quantification of the sensitivity would require the reproduction of the whole study. However, in most cases, the reflection index definition is not very sensitive. The optimization was required to distinguish between signal and noise.
We agree that the high precision of the reflection index does not make sense, so we reduced the number of digits after the comma.

*(12) Figs. 9–13: Please describe how the uncertainties are derived and what they mean. They are so large that they seem consistent with no reflections?*
According to (Gorbunov et al., 2006), the uncertainties are estimated as the widths of sliding radio holographic spectra, where the width in terms of Doppler frequency is transformed into the width in terms of the impact parameter.
Figures 9–13 indicate that the uncertainty in the case of no reflections are much larger for uncertainties in case of reflections. Therefore, the uncertainty estimates must be understood as a relative measure. This information has been added in the manuscript.

*(13) The RH formulation presented assumes spherical symmetry. What if the atmosphere is spherically symmetric but the surface is not? In Beyerle et al., there was discussion of how surface tilts will affect the reflected Doppler shift. Could you comment on how a surface tilt will affect your analysis?*
The presence of a surface tilt does not change the central idea of the approach. There is still a value of impact parameter $p_E$, for which a ray touches the tilted surface, and this value will be the boundary between the direct and reflected rays. Given the surface tilt $\alpha$ and ray incident elevation angle $\gamma$ with respect to the tilted surface, we can formulate the modified reflection law with respect to the unperturbed horizontal surface: the incident angle being $\gamma + \alpha$, the reflected elevation angle is $\gamma - \alpha$. The incident impact parameter being $p' = r_E n(r_E) \cos(\gamma + \alpha)$, the reflected impact parameter is $p'' = r_E n(r_E) \cos(\gamma - \alpha)$. The impact parameter perturbation at the surface is then equal to $p'' - p' \approx = -2 r_E n(r_E) \gamma \alpha$. In our examples, the maximum incident angle is about 0.01 rad, (which corresponds to impact parameter $p_E - 0.2$ km). If we assume that the tilt is about 0.001 rad (an estimate of the typical surface tilt of Antarctica), this will result in an impact parameter perturbation reaching a value of about 100 m.

On the other hand, the analysis of the influence of a realistic surface must involve the diffraction effects. This will help in the definition of the effective tilt as some sort of average over the Fresnel zone.

Minor corrections:

*(14) Throughout: "phase excess" should be replaced by "excess phase"*
This is corrected.

*(15) Above Eq. (24), p. 9: "$S_{MR}(r)$" -> "$S_{MR}(t)$"*
This is corrected.

*(16) p. 13: "None index" -> "No index"*
This is corrected.

[revised manuscript text omitted]
}_{\mathrm{Tx}}(t) \cdot \mathbf{u}_{\mathrm{Tx}}(t) - \mathbf{V}_{\mathrm{Rx}}(t) \cdot \mathbf{u}_{\mathrm{Rx}}(t) = c d_{MR}(t). \tag{24}$$

The phase excess is obtained by integrating the Doppler shift:

$$S_{MR}(t) = c \int \left( d^{(0)}(t) - d_{MR}(t) \right) dt, \tag{25}$$

where $d^{(0)}(t)$ is the vacuum Doppler shift for the direct rays, evaluated from (40), by inserting unit vector $\mathbf{u}_{\mathrm{Tx,Rx}}^{(0)}(t)$ corresponding to satellite-to-satellite straight-line direction. An example of reflected phase excess model is shown in Figure 7.

**3.2 Radio Holographic Index of Reflections**

The idea of flagging radio occultation with an index of the strength of the reflected ray consists in the following. Although the amplitude of the reflected signal is weak as compared to the direct signal, the instant frequencies of the reflected signal concentrate around the instant frequencies of the model reflected signal. Therefore, we can use the model reflected signal

[Figure]

**Figure 7.** Reflected phase excess model for occultation event 2008/01/01, UTC 01:02:23, 70.28°N 121.87°W (the same event as in Figure 2).

 $\underline{\exp\left(ikS_{MR}\left(t\right)\right)}$ as the reference signal and evaluate the radio holographic spectrum as follows:

$$\tilde{u}_R\left(\omega\right) = \int A\left(t\right)\exp\left(ik\left[S\left(t\right) - S_{MR}\left(t\right)\right] - i\omega t\right)dt. \underline{\int A\left(t\right)\exp\left(ik\left[S\left(t\right) - S_{MR}\left(t\right)\
[revised manuscript text omitted]